# Community-orientated primary health care: Exploring the interface between community health worker programmes, the health system and communities in South Africa

Hlologelo Malatji[1]*, Frances Griffiths[1,2], Jane Goudge[1]

**1** Centre for Health Policy, School of Public Health, Faculty of Health Sciences, University of the Witwatersrand, Johannesburg, South Africa, **2** Division of Health Sciences, Warwick Medical School, University of Warwick, Coventry, United Kingdom

* hlologelo.malatji@wits.ac.za

## Abstract

Due to insufficient number of health workers and the evidence of the benefits of community health workers (CHWs), CHWs are being deployed to provide health care services to under-served communities. In this article, we explore to what extent the South African CHW programmes introduced between 2009 and 2011 are attuned to community needs, integrated into the healthcare system and community structures, and also implemented in accordance with community-orientated primary health care principles. Using a case study approach, we studied CHW teams in seven primary healthcare facilities located in semi-urban and rural areas of Gauteng and Mpumalanga provinces, South Africa. We collected data using in-depth interviews involving facility managers, CHW supervisors, community representatives and key informants, and focus groups and observations of CHWs. The implementation of community-orientated health interventions remains complex. In the different sites, there were efforts to integrate the views of stakeholders (e.g., political leaders) into the implementation of the CHW programmes. However, many residents were more concerned about access to housing than health services. The CHWs services' were found to be generally comprehensive, however inefficient training, supervision and mentorship limited their effectiveness. The multidisciplinary approach to care, as introduced by some sites, helped enhance the knowledge and skills of some of the CHWs on complex health topics. The roll out of community orientated primary health care services is crucial in a resource-constrained setting like South Africa. However, significant socio-economic issues disrupt community involvement and the effective provision of services. Governments need to provide sufficient funds for training, supervision, supplies and remuneration to help overcome these barriers.

## Introduction

In response to the call for universal health coverage (UHC) and limited number of health workers, many low and middle-income countries (LMICs) are strengthening the delivery of primary

**Data Availability Statement:** Dataset is available on a reasonable request. Given the difficulties of anonymising qualitative data, we will work with

researchers wishing to further analyse the data. The request to access the dataset can be directed to Dr Olukemi Babalola, email: olukemi. babalola1@wits.ac.za.

**Funding:** The study was funded by Medical Research Council UK, DFID, ESRC and Wellcome Trust under the Joint Health Systems Research Initiative (JHSRI). The funders were not involved in the study design, data collection, analysis and preparation of this manuscript. All three authors received salary supplementation as part of the research funding from JHSRI We also want to acknowledge the support from the South African National Research Foundation (NRF) through the South African Research Chair Initiative (SARChi) in Health Systems and Policy at the Centre for Health Policy, University of the Witwatersrand.

**Competing interests:** The authors have declared that no competing interests exist.

**Abbreviations: CHW**, Community Health Worker; **COPC**, Community-orientated Primary Healthcare; **CRI**, Community Representative Interview; **FGD**, Focus Group Discussion; **KI**, Key Informant; **LMIC**, Low- and Middle-Income Country; **NGO**, Non-Government Organization; **PHC**, Primary Health Care; **UHC**, Universal Health Care; **WHO**, World Health Organization.

health care (PHC) services using community health workers (CHWs) [1, 2]. CHW programmes gained prominence post the Alma Ata Conference of 1978, which prioritised the strengthening of PHC to provide care to marginalised communities. Many large-scale CHW programmes often led by non-government organizations (NGOs) were introduced to provide the care in LMICs [1]. However due to insufficient investment and fragmentation, the performance of these programmes declined during the 1980s. However, there is now considerable evidence of the expansion and benefits of CHW programmes, particularly for the delivery of maternal and child health services [3, 4]. Reviews show CHW programmes need to be integrated with the community to be attuned to their needs, as well as integrated into the health services to be well resourced, well supervised, and to communicate their community's needs to practitioners and decision makers [5, 6]. There has been a growing interest in community-orientated primary care (COPC). COPC is an approach to health service development and delivery that integrates public health and primary care to deliver targeted services to a defined community [6]. This approach recognises the roles of lay health workers, civil society, non-governmental organisations and government sectors in the planning, designing and delivery of primary health care.

There are six key principles associated with the practice of COPC [5, 6]. Under COPC, there is a need to have a defined community, utilise multidisciplinary approach to care, use evidence to consistently evaluate and strengthen care, and the services need to be comprehensive and integrated into healthcare system and community structures (Table 1) [5]. Some countries such as Bolivia and Brazil have attempted to implement CHW programmes with a community orientation and in line with COPC principles. However, there are few evaluations of the effectiveness of these programmes [7] and often the initiatives do not embrace the complete COPC approach to primary healthcare development and delivery [6]. In South Africa, a national CHW programme has been implemented as part of strengthening PHC, and in some areas, implementers have tried to ensure that the CHW programme has been implemented with a focus on the COPC principles.

South Africa has a long history of COPC [8, 9]. In the 1940s, Drs Sydney and Emily Kark in Kwa-Zulu Natal province pioneered the first practice of COPC in Pholela Health Centre. Their multi-disciplinary approach to care consisted of clinicians, social services workers and lay health workers/ CHWs providing health care services to under-privileged communities. The model, though not widely implemented across the country, helped extend marginalised communities access to health care services. During the 1950s with the Apartheid government,

**Table 1. Six key principles of community orientated primary care.**

| Principle | Definition |
|---|---|
| A defined community | There should be understanding of the geographical location of the community and existing resources (e.g., churches and police stations) (assessment of local needs and assets). |
| Multidisciplinary approach to care | A team of health professionals should work in partnership with staff from social services to diagnose and respond to community problems. |
| Evidence based care | There should be consistent collection and use of community data to plan a service that will meet the community's needs. |
| Comprehensive care | The services package should provide care for a range of conditions, and where possible be comprehensive (e.g., promotive, preventative, curative and rehabilitation care). |
| Integration of care | Health care and other social services should be provided in an integrated and continuous way. |
| Community participation | Community members should be involved in the assessment of problems and implementation of the interventions, and where feasible existing community resources should be used (e.g., employment of local people as lay health workers). |

the COPC initiative came under increasing political and financial pressure which led to its demise in the 1960s [8].

Since the end of Apartheid and establishment of democracy in 1994, the provision of community-based health services has been led by NGOs in response to the HIV/AIDS and TB pandemics [10]. However, the NGO programmes were fragmented and single disease-focused, thus neglecting the other needs of individuals and families. In line with international UHC goals, in 2011, the South African Department of Health (DOH) introduced policies to strengthen PHC, and so to improve access to quality health care services [11]. A nation-wide CHW programme (locally known as ward-based outreach teams; WBOT) was introduced across all provinces, providing care for a more comprehensive range of health conditions [12].

The WBOTs comprise at least 6 CHWs, linked to a local PHC facility, serving a defined geographic area, providing a combination of promotive and preventative healthcare services to households, and making referrals to social services workers [13]. The WBOT team is meant to be led by an outreach team leader who is a nurse and supported by a health promoter and an environmental officer. The role of the outreach team leader is to provide field supervision to the CHWs during household visits, ensure the CHWs have the resources that they require (e.g. stationary) to fulfil their duties, and help establish team relationship with community structures. The CHWs receives training on identification of the need for antenatal and post-natal care, monitoring immunization and adherence to long-term medication, screening for malnutrition, TB, gender-based violence, making referrals to health and social services, and following up on patients who need to visit the health facility. The training is delivered in two phases, the first with a written examination (level 1), the second with a practical assessment in the field (level 2). At the time of the study, the CHW worked 6 hours per day, on a 12 months renewable contract earning a monthly stipend of R3 500 (201 USD).

Prior to the roll-out of the nation-wide programme, the majority of the CHWs worked for NGOs providing home-based care. They were absorbed into the national programme in 2011. Some districts in Gauteng province have taken a COPC approach to implementing the CHW programme. In Sedibeng and Johannesburg districts the COPC approach was established prior to the 2011 rollout; in Tshwane the COPC approach was adopted at the same time as the national rollout. More recently selected health districts in Western Cape and KwaZulu-Natal provinces have incorporated the COPC principles into their CHW programmes [9, 14]. In this paper, we examined whether the programmes as they existed at the time of fieldwork (2018–19 i.e 7–8 years post the nation-wide programme rollout) were being implemented in accordance with the principles of COPC.

## Methods

### Study design

Drawing on data collected as part of a larger CHW study [15] and doctoral research, we used a descriptive case study approach involving multiple qualitative methods to examine seven CHW teams. The use of case study design is suitable when researchers want to gain concrete, contextual and in-depth knowledge of a phenomenon. The design is further used to explore key features of a particular phenomenon. In the current study, the design enabled us to examine the practice of COPC in the context of the nation-wide CHW programme, CHWs relationship with the community and integration with the health system [16, 17].

### Study sites

We studied seven CHW teams in Sedibeng, Johannesburg, Tshwane and Ehlanzeni districts of Gauteng and Mpumalanga provinces. The distance between Gauteng and Mpumalanga

**Table 2. Description of site, CHW programme and COPC practices.**

| District | Location / Province | CHW programme prior roll out of national programme | Government WBOT programme | Principles of COPC formally articulated by programme leaders |
|---|---|---|---|---|
| Sedibeng | Peri-urban / Gauteng | Yes, implemented with COPC principles; no additional external funding | Yes, existing programme integrated into Gvt programme | Yes |
| Johannesburg | Urban / Gauteng | Yes, implemented with COPC principles; with additional external funding | Yes, existing programme integrated into Gvt programme | Yes |
| Tshwane | Urban / Gauteng | No | Yes, external funding that allowed additional COPC-related activities | Yes |
| Ehlanzeni | Rural / Mpumalanga | No | Yes, but CHWs remained employed by local NGOs | No |

facilities was approximately 500kms or 5hrs drive. These CHW teams and health districts were purposively selected based on their location, history and model of adopting the CHW programmes (Table 2).

The Sedibeng and Johannesburg districts adopted the COPC approach prior to 2011. The Tshwane district introduced COPC with the introduction of the national CHW programme in 2011. The Ehlanzeni district had not fully implemented the new CHW programme, nor did the leaders of the programme specifically focus on COPC. However, it was included in the study as the programme had reputation of being well run. We also anticipated that the differences between urban or semi-urban, and rural sites would be important, particularly in trying to deliver a community-orientated programme.

## Data collection

The data was collected by the first author, supported by a team of data collectors who were trained in PHC, community-based healthcare services, research methods and ethics. The first author was a male doctoral researcher employed by a research centre undertaking health policy and systems research. We conducted focus groups, observations, individual interviews and observations to gather data on the CHW programmes (Table 3). The data collection instruments were in English, however where participants struggled to understand the questions or express themselves in English, the team provided clarity using local languages (Isizulu, Sesotho and Sepulana, an unofficial language spoke in some parts of Mpumalanga province). Data were collected data from September 2016 to April 2019.

**Focus group discussions.** The data collectors facilitated nine focus group discussions with the CHWs recruited from the seven study sites. Every CHW who formed part of the CHW programme in these sites was invited to participate in the FGD. Where the CHWs

**Table 3. Data collection methods and number of participants.**

| | Sedibeng | | Johannesburg | Tshwane | | Ehlanzeni | | Total |
|---|---|---|---|---|---|---|---|---|
| | Team 1 | Team 2 | Team 3 | Team 4 | Team 5 | Team 6 | Team 7 | |
| Focus group discussions with CHWs | 1 | 1 | 2 | 2 | 2 | 1 | 1 | 9 |
| Observations with CHWs and their supervisors* | 24 | 16 | 19 | 7 | 5 | 10 | 17 | 98 |
| Interviews with CHW team | 4 | 4 | 3 | 2 | 1 | 1 | 2 | 17 |
| Interviews with community representatives | - | - | 5 | 0 | 4 | 6 | 5 | 20 |
| Interviews with key informants | 5 | | 2 | 6 | | 3 | | 16 |

Note:

* Each observation was a full day spent observing 2 or 3 CHW who were working together

exceeded the number suitable for a focus group discussion (7–10 participants), the researchers held two focus groups to accommodate the excess. There were no recorded refusals. Each FGD had approximately 10 participants, all women. Using a structured guide, we explored topics on the nature of their activities while out in the community, their experiences, successes and challenges while providing the services. Each of the discussions was audio-recorded, lasted approximately 2hrs, and was facilitated by the first author while data collectors helped take additional notes relating to the discussion.

**Observations.**   We designed observation templates and refined them using role-plays involving the data collectors. At each site, the data collectors observed a pair of CHWs for a period of 3 days. The CHWs were observed during household visits, the fieldworkers documented the services they provided and engagements with patients while in the field. Some of the observed visits were supervised by outreach team leaders (OTLs). However, in Ehlanzeni sites, the OTLs were not observed with CHWs, as the supervisors did not go to the households with the CHWs due to transport issues. The data collectors asked the household members for permission before carrying out the observations of the CHWs. There were no reported refusals to participate in the observations.

**Interviews.**   We conducted 17 semi-structured interviews with facility managers, OTLs and community representatives (CR). These participants were purposively selected based on their knowledge and experience of the CHW programmes in the different sites. We asked them about CHW duties in the households, their experience of CHW services, relationship with the healthcare system and community structures. We asked the OTLs and community representatives about the support they provide to the CHWs, the role they played in the community, successes and challenges of the programme. Each interview lasted approximately 45mins and were audio recorded with participant consent.

Furthermore, sixteen key informants (KI) drawn from Sedibeng, Johannesburg, Tshwane and Ehlanzeni districts were recruited using snow-ball sampling technique to participate in the study. The KIs occupied positions of family medicine practitioner, programme coordinator, nurse and academic/ researcher in the healthcare system, local municipalities and institutions of higher learning (Table 4). We asked questions about the origins of the programme, the practice of community-based health care, and the challenges of rolling out the programmes.

## Data analysis

We used inductive approach to identify themes emerging from the data, as well as framework analysis. The first author used a MS word sheet to extract data from the interview transcripts and observation notes from the seven study sites. The information was summarised while retaining important quotations. The author regularly presented the extracted data to the co-authors; this was done to ensure the completeness of the data and that all relevant information was being extracted. The resulting themes were group into broader COPC themes i.e. the nature of community-based health care services, the challenges CHWs encounter in providing services and evidence of COPC principles in the different sites. The themes were compared for similarities and differences in the different sites. The themes were synthesised to understand the history of the programmes, type of services provided by the CHWs, funding and socioeconomic issues of the communities. Moreover, evidence of COPC principles was also extrapolated from the different programmes.

## Ethics approval and participant consent

The larger project was cleared by the University of the Witwatersrand Medical Ethics Committee (M160354) and the Sedibeng health district. The doctoral study also received ethical

Table 4. CHWs, supervisors and key informant sociodemographic data.

| | | Sedibeng | | Johannesburg | Tshwane | | Ehlanzeni | |
|---|---|---|---|---|---|---|---|---|
| | | Team 1 | Team 2 | Team 3 | Team 4 | Team 5 | Team 6 | Team 7 |
| **CHW** | | | | | | | | |
| No. of CHWs | | 16 | 12 | 21 | 23 | 25 | 8 | 11 |
| Age (range) | | 26–53 | 26–51 | 26–42 | 28–47 | 29–50 | 30–58 | 36–61 |
| Years as CHW (range) | | 4–12 | 3–12 | 3–8 | 4–15 | 5–13 | 5–17 | 8–14 |
| No. of CHWs who have finished high school education | | 6 | 8 | 12 | 6 | 11 | 2 | 1 |
| **Supervisor** | | | | | | | | |
| No. of supervisors | | 3 (Professional nurse and enrolled nurses) | 2 (Professional nurse and enrolled nurse) | 2 (Enrolled nurses) | 2 (Enrolled nurse and Professional nurse) | 1 (Enrolled nurse) | 1 (Professional nurse) | 1 (Professional nurse) |
| Age (range) | | 43–65 | 25–59 | 35–37 | 29–42 | 45 | 35 | 55 |
| Years as nurse | | 14–36 | 3–39 | 4–5 | 4–9 | 7 | 10 | 17 |
| Years in the programme | | 0,6–4 | 0–4 | 4 | 4 | 5 | 0,2 | 5 |
| **Key informant** | | | | | | | | |
| Level | Academic/ researcher | - | | - | 2 | | - | |
| | Medical doctor | 2 | | 1 | 1 | | - | |
| | Nurse | 2 | | 1 | 2 | | 1 | |
| | Programme coordinator | 1 | | - | 1 | | 1 | |
| Institution | Government | 5 | | 2 | 3 | | 2 | |
| | University | - | | - | 3 | | - | |

clearance (M180140) from the same university ethics body, the Johannesburg, Tshwane and Ehlanzeni provincial government research authorities. The participants also provided written informed consent before they could participate in the study.

## Findings

In this section, we describe the CHWs, supervisors and key informants characteristics, history and implementation of CHW programmes in the different sites. We then identify and describe evidence of the extent of community orientation of the programme, using the COPC principles as described in Table 1.

### CHWs, supervisors and key informants characteristics

The CHWs were aged between 26 to 61 years. The least experienced CHW had 3 years of service, while the longest serving CHW had 17 years of services. The majority of the CHWs had not completed their high school education, Teams reporting low number of CHWs with high school education where in Ehlanzeni and Tshwane districts (Table 4).

In terms of supervision, the sites made use of enrolled nurses and professional nurses. The Sedibeng sites used professional nurses and enrolled nurses, while in Tshwane there was a similar supervision model as Sedibeng for one site, the other site used only enrolled nurse (Team 5). Ehlanzeni sites used only professional nurses to provide supervision to the CHWs. The supervisors were aged between 29 to 65 years, and have worked as nurses in different institutions for 4 to 39 years. At the time of the study, the supervisors had been serving as CHW supervisors for 6 months to 5 years.

### The CHW programmes and the communities they serve

**Sedibeng.**   Due to increasing cases of chronic diseases and shortage of health workers to provide health care, a community-based health programme was introduced in 2010. Through the assistance of local gatekeepers (i.e., political leaders), health posts (HPs; temporary wooden structures) were introduced in one of the more densely populated areas of Sedibeng district. Nurses and lay health workers (CHWs), selected from local primary health facilities and NGOs respectively, were recruited to work on the programme. The CHWs received training from a team of family medicine specialists to undertake a community diagnosis, involving the collection of individual and household data (e.g., number of householders, income, access to social assistance, self-reported illness etc.), to identify health, social and economic needs.

An unpublished evaluation by members of Sedibeng health team indicated an increase in the number of identified TB, hypertension and child malnutrition cases (Table 5). However, with the introduction of the nation-wide CHW programme in 2011, the programme was experiencing challenges, as funds and other institutional resources were diverted to the nation-wide CHW programme.

At the time of the study, Team 1 provided services to an informal settlement, shacks made from corrugated iron and plastic, were home to local South Africans and immigrants (Table 5). Access to running water, sanitation and electricity was limited. The residents relied on water taps installed in central points of the community. The two CHW teams were operating from a health post and PHC facility and led by junior and retired senior nurses. The CHWs provided range of promotive and preventative services including delivering chronic medication to pensioners. The CHWs, using standardized forms, undertook household registration, contact tracing and made referrals to local PHC facilities. However, the CHWs stored the forms in their homes and the information wasn't used to inform the type of care provided to the households.

**Johannesburg.**   In Johannesburg, community orientated care was introduced by family medicine practitioners in one community health centre prior to 2011. The practice was called

**Table 5.  Descriptions of the CHW programmes and the communities they serve.**

| | Sedibeng (semi-urban) | Johannesburg (urban) | Tshwane (semi-urban) | Ehlanzeni (rural) |
|---|---|---|---|---|
| **Programme inception** | • Introduced in 2010 prior to the nation-wide programme<br>• Funded by district management | • Introduced prior to 2011 by family medicine practitioners<br>• Externally funded | • Introduced in 2011 by family medicine practitioners & local university<br>• Externally funded | • Nation-wide programme<br>• Programme partly managed by govt and NGOs |
| **Supervision** | Professional and enrolled nurses | Enrolled nurses | Professional and enrolled nurses | Professional nurses |
| **Housing and infrastructure** | • RDP<br>• Formal brick housing<br>• Informal settlements with unreliable electricity and water supply | • Formal brick housing<br>• Hostels and informal settlements with unreliable electricity and water supply | • RDP<br>• Formal brick housing<br>• Informal settlement with unreliable water supply<br>• Poor roads | • Formal brick housing<br>• Unreliable water supply and poor roads |
| **Perceived impact of services** | • Increase in identified TB, hypertension and child malnutrition cases | • Improved chronic care management | • Improved chronic care management | • Improved access to general healthcare |
| **Sustainability** | • Funds diverted to the nation-wide CHW programme | • External funding come to an end<br>• Integrated into the nation-wide programme | • External funding ended<br>• Implementation weakened | • National programme only<br>• On-going |

Chiawelo Community Practice and operated within the community health centre. At the time of the study, it also functioned as one of the local universities teaching and research site. Similar to the Sedibeng team, there were community consultations to gain permission to work in the community. The catchment area was mapped, and community interviews carried out by CHWs to understand community needs. The team registered 22,000 households. All the registered householders were told about the community practice (e.g what it intend to achieve and how they will benefit). Patient care was provided by a medical doctor, clinical associate, nurse and a team of eleven CHWs recruited for the practice. The services were limited to the individuals registered under the community practice.

After several years, the initiative stalled as external funding used to set up the practice had come to an end. The CHWs were absorbed into the national CHW programme, resulting in contractual conflicts between the CHWs and programme leaders, as their integration into the national programme meant a reduction in their monthly stipend. At the time of the study, no evaluations had been published on the outcome of the practice; anecdotal evidence suggested an improvement in chronic care management.

This team worked in a diverse area with formal brick housing, hostels (originally built for working miners, but now occupied by families), as well as informal settlements. Communities with formal housing tended to have access to running water and electricity and were relatively affluent; the hostels and informal settlements were not (Table 5). High levels of crime made it difficult for the CHWs to adequately provide care. A CHW commented "*Many criminals live in this hostel. We are always scared when walking in this area. It is worse because we have to enter the households and provide care*" (CHW, FGD, Team 3). The CHWs preferred to visit their hostel clients in groups, however, this was not always possible; to reach their allocated number of patients they had to split up.

**Tshwane.** COPC informed intervention was first introduced in Tshwane as part of local government health services, with a group family medicine practitioners and researchers leading the initiative. In 2011, the initiative became part of the nation-wide CHW programmes with external funding. Similar to the Sedibeng district, in Tshwane local schools and churches were used as health posts, particularly in areas where there were few PHC facilities. In rolling out programme, the CHWs mapped the community and undertook a community diagnosis. The CHWs were equipped with tablets with a specifically designed app to gather household and individual data. Data from CHWs' tablets was aggregated by the supervisor and programme leaders for supervision and in-service training purposes.

The programme implementation relied on the CHWs employed by the Department of Health, as part of the nation-wide CHW programme. The CHWs had only been offered 10-day training by the Department and the majority had low literacy levels. The low stipend, lack of working tools and low literacy levels led to a conflict between the CHWs and the Department of Health, and meant the CHWs were often unreliable pool of community workers. Similar to the other health districts that had attempted to implement community-based health services in accordance with COPC principles, implementation of the programme weakened when external funding came to an end.

The two teams were providing services to residents in informal settlement, RDP (These are houses that have been built by the government and are given to low-income families, as part of the Reconstruction and Development Programme introduced in 1994) and formal brick housing. The informal settlements were without running water or electricity, and the majority of the residents were unemployed. In some part of the community, there was sewage in the streets making the area inaccessible by foot and motor vehicle. The CHWs prioritized providing care to patients located in the informal settlements, as many of the residents faced challenges in

maintaining medication protocols for chronic conditions. Ensuring patients took their medication as prescribed required frequent visits (Field observation notes, Team 4).

**Ehlanzeni.** The national CHW programme was introduced in the district in 2011. CHWs, who remained in the employment of the NGOs, were selected to work on the new programme (Table 5). Their monthly stipend was paid by the provincial Department of Health via the NGO. The two CHW teams were overseen by senior professional nurses based in the local PHC facilities; the CHW also reported to program managers attached to the NGOs. The CHWs undertook a combination of health promotion and preventative roles in the surrounding rural communities. While the CHWs had prior work experience in providing home-based care services, none of the CHWs had received CHW training prior to being inducted into the CHW programmes. The appointed supervisors were unable to provide field support to the CHWs due to lack of transport to accompany the CHWs while visiting households. The supervisors spent most of their time in the facility attending to patients.

In the communities, many households had an unreliable water supply. The majority of the residents didn't have water boreholes and had to use wheelbarrows loaded with 20 liters containers to ferry water from households with boreholes. The area had poor roads, inaccessible by a motor vehicle (Table 5). Pensioners often struggled to visit the healthcare facility, as they could not walk or get a lift (Observation notes, Team 6). The CHWs complained about the hindrance of local traditional beliefs in their efforts to provide care. For example, during household visits, the researchers observed a diabetic pensioner with a wound on her foot refusing CHW care; she believed her neighbour was the cause of her poor health and misfortune (Observation notes, Team 6). As local women, the CHWs had to compromise and accommodate the patient beliefs.

## Community orientation of the CHW programmes

The CHW programmes demonstrated different degrees of the key features of community-orientated primary health care (i.e., analysis of local health needs and assets, use of evidence, service integration, comprehensive care and multi-disciplinary approach to care, and community participation).

**A defined community and use of evidence.** The Tshwane, Sedibeng and Johannesburg teams gathered data to understand the local context and health needs. CHWs visited households to learn about their perceived needs. Key informant in Sedibeng commented: "*The community participated by identifying its needs, the needs identified by the community were then moderated to come up with a list of urgent needs to be addressed*" (KI, interview, Sedibeng). Undertaking a community diagnosis also allowed the implementing teams to explain the programme to the community and involve the community. The data generated through this process was collated and used to inform the human resources and service requirements for the intervention.

Across the different sites, there was evidence of continuous data collection to inform service delivery. In Tshwane district, as mentioned above, a mobile technology was installed in tablets given to CHWs to records householders' health needs, care provided and schedule follow-up visits. However, due to limited funding, the use of mobile health technology could not be sustained, as lost and malfunctioning tablets were not replaced or repaired. A key informant commented: "*unfortunately, if a technology device is used every day, those devices last for about 2 years. Therefore, you need to budget for a 2-year replacement cycle. Otherwise, it doesn't really work*" (KI, Interview, Tshwane). Key informants and CHWs mentioned the Department of Health did not replace or refurbish lost or damaged tablets. The CHWs who damaged or lost

the tablets regressed to the traditional way of collecting and recording patient data on a piece of paper.

In the other districts, there was no use of mobile technology. Instead, household information was recorded on paper, often filed at the CHW homes, and not used to help inform future interventions in the community. However, some of the informants were concerned that under the national CHW programme the use of evidence to guide practice is diminishing. A key informant in Sedibeng commented: "*. . .. WBOT asks you questions regarding your needs, but only God knows for what, because I haven't seen anyone acting on the collected data. . . WBOT is designed not to have impact on the health of the people*" (KI, Interview, Sedibeng).

**Service integration.** To assess CHWs connection with the health system, we examined four features of the programmes—distance/ location from the clinic, supervision, relationships with health workers and referral systems.

*Distance / Location*. In the rural setting, many villages were located far from health facilities. To reduce the time spent walking between their homes, the health facility and the community, the CHW were advised to only report to the facility on Fridays to compile their weekly reports. On the other days of the week, they went straight from their homes to visit households: "*They realized some of us come from far places, so coming to the clinic just to sign in and later go back to the households was impractical.*" (CHW, FGD, Team 7). Though this change helped save the CHWs time, it reduced their access to supervision from the facility-based nurses/supervisors. In contrast, in urban sites, the catchment areas were conveniently located closer to the health facilities, and although supervisors lacked transport, they were able to walk or used their own cars.

*Supervision*. When CHW teams were led by senior nurses who provided thorough supervision, the CHW were more competent, likely to receive support from the facility-based staff members and be respected by the community. More skilled, senior supervisors discussed complex cases with CHWs (e.g., patients refusing care and not adhering to treatment) and where necessary accompanied the CHW to see the patient. However, this was not the case in the rural teams, where, although the supervisors were senior nurses, it was not possible for them to invest sufficient time and resources in the supervision of the CHWs. Some of the CHWs in Ehlanzeni were without a supervisor for a prolonged period during data collection. "*we have not had a supervisor since October last year. . . when we have problems we tell the facility manager who makes time to assist us*" (CHW, FGD, Team 6). During data collection, a supervisor was appointed and had to undergo an induction by the Department of Health. The other supervisor in Team 7 only saw the CHWs when they reported to the facility on Fridays; during the week she didn't have a vehicle to visit households. The CHWs were allowed to call the supervisor to discuss issues they encountered. However, this was impractical as they were not provided with airtime and couldn't afford buying airtime for work purposes. The contracting NGO was also not providing daily supervision to the CHWs.

*Relationships with health care workers*. The CHWs in Team 2, led by a senior supervisor, received sufficient support from facility-based staff members, as the supervisor negotiated for their inclusion in facility activities and borrowed equipment (e.g., blood pressure machines) on their behalf. In order to build a good relationship with facility staff, the supervisors in Johannesburg helped the nurses in the clinic: "*to ensure the patients referred by my people (CHWs) are attended timeously, I make time to assist the nurses in the clinic. I sometimes assist X in the triage room or attend women who are visiting the clinic for family planning services*" (CHW supervisor, interview, Team 3). However, support for CHWs planned activities was not forthcoming: "*I usually go to the facility manager's office and inform her that I am having a campaign based on HIV, TB, vitamin A or de-worming. She will be happy but there will be no*

*assistance, even if I can ask for assistance from the EPI or TB side. They will say, you are independent, you must know these things*" (CHW supervisor, interview, Team 5).

In the rural sites, when the CHW were at the clinic, they used a separate building, not the main clinic; this arrangement isolated the CHWs from the rest of the facility staff members. The team leaders also spent less time with the CHWs, as they only visited the facilities on Fridays.

In the urban sites, the CHWs had a complex work relationship with facility-based health workers. In one of the facilities, nurses used degrading names to refer to the CHW. A CHW commented: "*they call us names like "mamoroto" [someone who is working with urine]... They always chase us around saying we must go to the side where patients' urine is collected*" (CHW, FGD, Team 5). This degrading name was conceived from the CHWs role in the collection of patients' urine as part of their duties within the facility.

*Referral systems*. CHWs refer cases they cannot resolve at household level to health facilities for further assessment and care. A referral requires cooperation between the CHWs and facility staff (i.e., nurses providing feedback to the CHWs). However, due to CHWs "inferior" status, some nurses did not complete the back referrals, thus hindering CHWs from providing care post referral. A CHW commented: "*we always have to beg to receive the report about the patients that we referred to the clinic*". Another CHW added: "*getting the completed back referral form from the nurses is never easy, I always have to ask my supervisor to get the form from the clinic staff*" (CHW, FGD, Team 1). These issues were more common in teams where the CHWs did not have a senior or present supervisor to mediate between them and facility staff.

**Comprehensive and multi-disciplinary approach to care.** In exploring the features of comprehensive care and use of multidisciplinary teams to deliver care, the use of experts to provide advice and mentorship, CHWs limited knowledge and skills, and unmet health needs emerged as subthemes.

*Expert advice and mentorship*. In the urban sites, the CHWs received expert advice and mentorship from clinicians (e.g. medical doctors) and social services professionals in order to provide health promotion, prevention, screening services and referral for a wide range of health conditions. In Tshwane district, the CHWs met with members of a multi-disciplinary team weekly to debrief and receive case guidance. A key informant commented: "*Last week, I attended a meeting in the X clinic where the CHWs and OTL had a review meeting. The multi-disciplinary team consisting of a dietician, doctors, nurses and so forth attended the meeting. They reviewed the difficult cases and advised the CHWs accordingly*" (KI, interview, Tshwane). Similarly, the Sedibeng and Johannesburg teams had the support of medical doctors, nurses and clinical associates. The multi-disciplinary approach benefited the CHWs, as they had access to clinical knowledge shared during the meetings. In these meetings, the CHWs presented complex cases they dealt with in the field and received expert advice or mentorship.

*Limited knowledge and skills*. However, there were concerns that the CHWs did not possess the knowledge and skills required to implement support interventions aligned to COPC. A key informant in Tshwane district commented: "*you cannot expect people who are at that level of training, who are not managed properly and without team leaders to function well*". To overcome this challenge, for example, the Johannesburg team used external funds to employ, train and retain CHWs with desired knowledge and skills mix. In Sedibeng, the CHWs experienced numerous challenges when offering services to those without residency status: "*Working in this community is never easy for us. The CHWs do visit the householders to provide services, however some the householders do not have identity documents needed to register them for social services*" (CR, interview, Team 1). The CHWs reported these cases to their senior team leaders who couldn't assist as they did not go to the fields with them to offer onsite supervision.

*Unmet health needs*. The identification of unmet need also appeared to present challenges for the healthcare system. The Sedibeng team during household visits found suspected cases of cervical cancer, and those who needed cataract operations. However, care for such conditions was only available at distant, tertiary hospitals, and required numerous visits. The majority of the householders who received care from the CHWs were pensioners and unemployed women with dependents. They did not have the funds to visit the facilities. As a result, some of the participants deemed the attempt at COPC informed intervention as ineffective, as the identification of community health needs was not matched with available care.

**Community participation.** One of the key principles of COPC is for health programme developers to engage communities and ensure the services to be provided are relevant to the needs of the community. It is important that there are open channels of communication such that if there is a misunderstanding, issues can be clarified.

*Setting up programmes*. Each programme consulted local stakeholders such as political and traditional leaders, school managers and local NGOs. In Tshwane, the team worked with local stakeholders to secure space in their schools and churches to be used as health posts. A participant commented: "*After we have introduced and involved the community in our work, implementation went well. The ward councillor even helped us to find a suitable space to use as a health post*" (KI, interview, Tshwane). Another participant added: "*Since we informed them about our programme. We were protected, nothing bad has happened to us because the community know us. If I complain that somebody snatched my phone, they will make sure that they will get whoever took it*" (KI, interview, Tshwane). In Johannesburg at the start of the programme, there was resistance from patients and community, but continuous engagement led by a community liaison officer who was employed to support the programme helped ease the tension between the health team and community. Similarly, in Sedibeng (Team 1), local leaders upon realizing patients visiting the health post queued in the sun or cold weather, volunteered their time to build a temporary shelter for the patients to use while waiting for their turn.

*Maintaining on-going relationships with service users*. In Sedibeng, when the CHW team 1 struggled to persuade patients to visit the local health facility for further health assessment and treatment, community leaders made time to meet and listen to the CHWs problems (e.g., patients refusing CHWs entry into their households), and addressed the issues with the householders. In the different sites, the CHWs also struggled to access some households to provide care due to the stigma associated with their services. Due to their previous role during the height of the HIV pandemic, their service was associated with providing HIV and TB care. A CHW commented: "*..the clinic sometimes send us to clients who are defaulting on their medication, but because they [clients] don't want to be seen being visited by us, they chase us away*" (CHW, FGD, Team 3).

*Engagement with the broader community*. In Ehlanzeni (Team 6), a community representative who also served as a member of the clinic committee ensured the clinic was always represented during community meetings she addressed. This leader's initiative helped ensure service delivery and health concerns of the residents were discussed in the same platform. However, community members were only interested in topics such as access to employment, water and decent housing. Many of the communities were politically volatile, with residents were regularly protesting for decent housing and running water. In Sedibeng district, a health post was burned down by members of the community protesting against continuous electricity cuts in the area, while in another unrelated event district health officials were held hostage by CHWs protesting against poor work conditions (e.g. low stipend). These on-going service delivery protests often disrupted the engagements between the health team and community: "*We often have to stop work, because of service delivery protests*" (KI, Tshwane).

## Discussion

Our study examined whether the community-based health programmes as they existed prior and after the nation-wide CHW programme were implemented in accordance with the COPC principles. To summarise our findings, we found some evidence of resistance by community members to participate in the design and implementation of the health programmes, as the residents were more interested in other services such as housing. Also, CHWs insufficient health knowledge and skills contributed to the ineffectiveness of the programmes, particularly where the CHWs did not have access to a senior team leader to provide them with supervision and mentorship. However, the multidisciplinary approach to care which saw medical doctors providing in-service training to CHWs in some sites, improved the services provided to patients. As much as there was an attempt to implement the programmes in accordance with COPC principles, dysfunctional mobile phones for patient data collection, volatile communities and an unsupportive healthcare system limited the effectiveness of the programmes to deliver care. The CHW teams had to navigate these issues to deliver healthcare to residents.

Studies have shown the delivery of community orientated health care requires well-structured support from healthcare systems [18]. A review of literature in LMICs has found many lay health workers did not possess the training and knowledge to provide maternal and child health services, however with continuous clinical support they were able to acquire the knowledge and learn skills to provide quality care [19]. A study in Brazil found community health agents who worked as part of a local family healthcare team, which included doctors and dentists, to be sufficiently supported due to their integration into healthcare systems and community structures [20]. In our study, some programme leaders organised in-service training opportunities for the CHWs to enable them to provide comprehensive care to their clients. However, in our rural sites, contextual factors such as distance and lack of transport often limited the availability of supervision, the CHWs functioned without field supervision. The CHWs who hardly interacted with facility-based workers received less support in their daily functions. The unavailability of senior team members in some sites meant conflicts the CHWs had with members of staff were left unresolved.

CHW programmes can fail due to over-reliance on external funds to set up and implement the programme [5, 6, 18]. For example, in Kenya and Brazil, initiatives stalled when donor funds ran out, and the national governments did not make funds available to continue the initiatives [6, 18]. In our study, the programmes studied relied heavily on external resources to recruit, train and retain the CHWs, and unfortunately, when the funds ran out the initiatives stalled. Many of those writing about community-based programmes argue sustainable initiatives require financial commitment from government [21].

World Health Organisation guidelines states communities have a significant role to play in CHW selection, programme implementation, supervision and performance evaluation [2, 6]. However, a review of CHW programmes has shown communities are poorly involved in setting up health programmes, mainly due to external forces which include pressure from donors and technical advisors to achieve quick results, thereby bypassing the slow social processes required to establish stronger ties between the CHWs and communities [18]. In our study sites, the communities were not involved in CHW selection, programme implementation, supervision or evaluation. The efforts of programme leaders to consult with community members were not met with the same energy. The majority of communities had housing issues, and as a result were less invested in health programmes. The implementing teams had to navigate the community unresponsiveness while delivering healthcare services.

In response to the limitations of CHW programmes, Schneider and Lehmann propose efforts be directed to develop community health systems [22]. The current organisation of

CHW programmes are ineffective as they are narrowly focused using CHWs as key drivers of health service delivery. Other sets of actors and systems within the community, that can be tapped into and positioned to improve the delivery and uptake of health services are being neglected and underutilised [22]. In their work on community health systems, Schneider et al recognise the role of non-government organisations, civil society and government sectors (e.g education and social development) in designing and delivering care. Noting countries under investment in CHW programmes, the scholars recommend governments allocate sufficient resources to the health sector for training, supervision, reliable supplies, improved data systems and integration of the programmes into health systems and community structures [23].

The study had several limitations and strengths, first, the CHWs and their supervisors, and the key informants may have provided socially desirable answers in favour of their unique COPC initiatives. To mitigate this, data were collected from different participants located in the different levels of the healthcare systems (e.g. program directors vs. CHWs). This approach allowed us to obtain multiple perspectives of the initiatives and triangulate the data. Second, there isn't a single study on the practice of COPC that has attempted to quantify the principles, this make it difficult to objectively assess whether the implementation of CHW programmes is in line with the COPC principles or not. Third, the data collection tools were in English, this might have resulted in different explanation and interpretation of the interview questions during data collection. The study also demonstrated some key strengths. As far as we are aware, there are limited studies locally that explored the application of the COPC approach in evaluating the design and delivery of community-based care under the nation-wide CHW programme. Existing literature, studies the COPC programmes in isolation [9, 24, 25], thus missing the opportunity to highlight the key design features that can be used to strengthen nation-wide CHW programmes in LMICs.

## Conclusion

The implementation of community-orientated primary health care services is crucial in a resource constrained setting like South Africa. However, the implementation of these programmes often get derailed by socio-economic issues dominant in the communities. Poor community participation in the design and implementation of the programmes, limited health system support for outreach teams and unsustainable funds make it difficult for the teams to be effective in the delivery of care. In order to strengthen these programmes, CHWs need to be integrated into healthcare systems so to access supportive supervision, have access to resources, and governments make available funding to implementing teams.

## Supporting information

**S1 Checklist. COREQ (Consolidated criteria for Reporting Qualitative research) checklist.** (DOCX)

## Acknowledgments

We would like to thank the following individuals for their invaluable contribution: the data collectors, key informants, CHWs and their supervisors, health facility staff members and community representatives. The support of the Sedibeng Health District management, in particular the former district director Mrs. Salamina Hlahane and community-based services coordinator Mrs. Bridget Lefhoedi is highly appreciated.

We would also like to thank these organisations for their generous support: Sedibeng, Johannesburg, Tshwane and Ehlanzeni Health Districts, and the UK Medical Research Council (MRC).

## Author Contributions

**Conceptualization:** Frances Griffiths, Jane Goudge.

**Data curation:** Hlologelo Malatji.

**Formal analysis:** Hlologelo Malatji, Jane Goudge.

**Funding acquisition:** Jane Goudge.

**Investigation:** Frances Griffiths, Jane Goudge.

**Methodology:** Frances Griffiths, Jane Goudge.

**Project administration:** Hlologelo Malatji.

**Resources:** Frances Griffiths, Jane Goudge.

**Supervision:** Hlologelo Malatji, Frances Griffiths.

**Validation:** Jane Goudge.

**Writing – original draft:** Hlologelo Malatji.

**Writing – review & editing:** Frances Griffiths, Jane Goudge.

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
