## [Decision Letter · Decision Letter 0]

23 Aug 2022

PGPH-D-22-01143

Community-orientated primary health care: exploring the interface between community health worker programmes, the health system and communities in South Africa

Dear Dr. Malatji,

Thank you for submitting your manuscript to PLOS Global Public Health. After careful consideration, we feel that it has merit but does not fully meet PLOS Global Public Health’s publication criteria as it currently stands. Therefore, we invite you to submit a revised version of the manuscript that addresses the points raised during the review process.

Please see the comments from the reviewers below. We were fortunate have multiple reviewers and some of the concerns have been expressed by more than one reviewer. 

We look forward to receiving your revised manuscript.

Kind regards,

Shailendra Prasad, MD, MPH

Academic Editor

Journal Requirements:

a. State what role the funders took in the study. If the funders had no role in your study, please state: “The funders had no role in study design, data collection and analysis, decision to publish, or preparation of the manuscript.”

b. If any authors received a salary from any of your funders, please state which authors and which funders.

2. Please ensure that your Financial disclosure statement is matched with the funding information.

3. In the online submission form you indicate that your data is not available for proprietary reasons and have provided a contact point for accessing this data. Please note that your current contact point is a co-author on this manuscript. According to our Data Policy, the contact point must not be an author on the manuscript and must be a third party. Please revise your data statement to a non-author institutional point of contact, such as a data access or ethics committee, and send this to us via return email. Please also include contact information for the third party organization, and please include the full citation of where the data can be found.

4. We have amended your Competing Interest statement to comply with journal style. We kindly ask that you double check the statement and let us know if anything is incorrect. 

Additional Editor Comments (if provided):

Reviewers' comments:

Reviewer's Responses to Questions

**Comments to the Author**

1. Does this manuscript meet PLOS Global Public Health’s publication criteria? Is the manuscript technically sound, and do the data support the conclusions? The manuscript must describe methodologically and ethically rigorous research with conclusions that are appropriately drawn based on the data presented.

Reviewer #1: Yes

Reviewer #2: Yes

Reviewer #3: Partly

Reviewer #4: Partly

Reviewer #5: Partly

Reviewer #6: Yes

2. Has the statistical analysis been performed appropriately and rigorously?

Reviewer #1: N/A

Reviewer #2: N/A

Reviewer #3: N/A

Reviewer #4: N/A

Reviewer #5: N/A

Reviewer #6: N/A

3. Have the authors made all data underlying the findings in their manuscript fully available (please refer to the Data Availability Statement at the start of the manuscript PDF file)?

Reviewer #1: No

Reviewer #2: Yes

Reviewer #3: Yes

Reviewer #4: No

Reviewer #5: No

Reviewer #6: Yes

4. Is the manuscript presented in an intelligible fashion and written in standard English?

Reviewer #1: Yes

Reviewer #2: Yes

Reviewer #3: Yes

Reviewer #4: Yes

Reviewer #5: Yes

Reviewer #6: Yes

5. Review Comments to the Author

Reviewer #1: There are several confusions that the authors must clarify. 1) When the authors write , "However, many residents were more concerned about access to housing that health services delivery issues" I am not sure whether the authors are referring to the patients of the CHW. In the sentence quoted above do they mean, access to housing than health services. If yes, then what is the implication of this? That CHW were not perceived as important on the field by the residents? IF so, then this must be emphasized as the gap between consensus in public health about the importance of CHWs and service recipients who see other public goods as important such as housing.

When the authors write, "Governments need to provide sufficient funds for training, supervision, supplies and remuneration to help overcome these barriers" they are not paying heed to the history of CHWs. It was precisely because of various lacks and scarcities were CHWs seen as important. Are there are cost analysis studies being done? For example, look at Sarang Deo's work.

Finally, please look at the work of Svea Closser, Alex Nading, on the kind of work CHWs do and the kind of figure they are in the structures of national public health.

Reviewer #2: Thank you, editor, for providing the opportunity to review this paper. This reviver has the following observation for consideration in this paper.

1. Abstract – the abstract seems more general. Authors can write more specifically about south Africa from the beginning, including the multidisciplinary approach and the socioeconomic barriers are?

2. The authors seem more focussed on COPC as the paper's main argument, therefore mentioned in the title. The current title looks more methodological paper. That is clearly evident in the framing of the introduction. Authors can reframe the introduction section with the beginning of the CHWs programs (specific) and the South African health system context, where some research gaps can be justified. Authors can address those research gaps using the COPC approach as a method, including other methods used in the research.

3. As the authors mentioned as a case study as a methodological design, some explanation of how it is case study design is appropriate for this could be important to justify the methodological justification.

4. Regarding the data collection and numbers of participants, the authors might have selected them using information saturation, which can be explained to demonstrate why the authors selected specific numbers of participants.

5. Data analysis is unclear whether it is thematic analysis or framework analysis. The authors indeed used a deductive approach to fit contents in the COPC framework. Thus, more clarity is needed here.

6. The first part ( site-specific explanation of findings )of the result section is redundant with the latter part of the findings. The first part can be summarised in the table with a key description.

7. It is unclear what themes emerged in the data analysis in each COPC component and what supporting evidence for these themes. The authors provided a long text of paragraphs, including verbatim and descriptions. Authors can write themes under each COPC component. That might be more structured and explanatory. More analysis of the data is needed.

8. There is good content in the discussion section; however, the writing structure made it difficult to follow. In the first para of the discussion section, the authors can summarise key overarching findings that can be discussed in subsequent paragraphs using the similar framework used in the finding section. So, from the contents and structure point of view, this section required more revision to draw the key implications for the policy and programs regarding CHWs programs in SA.

9. Authors can include limitations of the study and section programs for policy and programs too.

10. The current conclusion section is general. Authors can include key points. The first line of the conclusion is your method/approach, not conclusions. The conclusion should be made based on the findings discussed in the paper. More specific conclusions are needed.

I hope the above points will help to improve the paper.

Reviewer #3: Thank you for inviting me to review this important study. There are some issues regarding the clarity of objective and research methodology that could help improve the paper

In the methods, the authors state that they used “a descriptive case study approach”. Could the authors please give a bit more details so that it sets the reader up a bit better?. Perhaps stating clearly that multiple qualitative methods will be used?.

The objective to examine “the extent” of community orientation of program is also a bit vague. How can the “extent” of quantified or summarized?

The clarity of quantitative analysis could improve. As written, the authors have used the COPC principles as their main headings. Some headings have themes but some do not. For example for the heading “service integration” there are 4 themes (distance/location, supervision, relationships, and referral systems) but for example, for “comprehensive and multi-disciplinary care” and “community participation”, there are no themes, but based on the descriptive given, some themes may have emerged.

Along with my previous comment, please visit the COREQ checklist or SQRQ checklist so that all qualitative components are adequately described and summarized.

I could not access the observation template, but as there were close to 100 observations, perhaps some of the observations made can be summarized quantitatively.

Reviewer #4: This paper titled “Community-orientated primary health care: exploring the interface between community health worker programmes, the health system and communities in South Africa” reports a 6 year long qualitative study that evaluates the implementation of various community health worker programs in 4 districts of South Africa as to how well-integrated the program is with the communities and how they follow COPC principles.

The authors have looked at community health workers through the lens of community oriented primary healthcare and evaluated how well the CHW programs are implemented as measured against the ideals of primary healthcare. To do this, with support from a team of data collectors, the first author interacted with people at various levels - CHWs, facility managers, CHW team leaders, and community representatives. They also observed the CHWs at work for 3 days each. To get a wider perspective they also interacted with key informants from district departments of health, local municipalities and institutions of higher learning. Overall a wide variety of individuals have contributed to the data collection process. This leads to very nuanced observations, like “WBOT asks you questions regarding your needs, but only God knows for what because I haven’t seen anyone acting on the collected data... WBOT is designed not to have impact on the health of the people”. This demonstrates that the team was able to break through diplomacy and arrive at raw observations.

Study sites

Although there is clarity regarding the diversity of the 4 districts selected, it is unclear as to why these 4 districts were selected. It appears like there are 52 districts in South Africa, and these 4 districts (Sedibeng, Johannesburg, Tshwane and Ehlanzeni) seem to be geographically close to each other. This might have implications on extrapolating the experience to the nation-wide program.

Further, it isn’t clear whether the government program has three different models. Table 2 talks about “Gvt programme; no additional funding”, “Gvt programme with additional external funding”, and “Gvt funding but sub-contracted to local NGOs”. Later in the discussion section, 464-468 lines: “In our study, the COPC initiatives studied relied heavily on external resources to recruit, train and retain the CHWs, and unfortunately, when the funds ran out the initiatives stalled. Many of those writing about community-based programme argue sustainable initiatives require financial commitment from government”. Is the analysis about the nation-wide program or on the different CHW programs that has been running in the 4 districts?

This question on what is being evaluated made me, as a reader, confused when reading the “Findings” section. For example in 204-206 lines, the authors say “However, with the introduction of the nation-wide CHW programme in 2011, the programme was experiencing challenges, as funds and other institutional resources were diverted to the nation-wide CHW programme”. Which program was observed by the team in Sedibeng? The nation-wide CHW program or the one that was introduced in 2010? Which one is the analysis about? Did the programs get merged in 2011? As this reviewer is not well-read on CHW programs in South Africa, it would help to have a clarifying explanation.

Similar confusion occurs to this reviewer on reading the history in Johannesburg.

It could be that the story of origin of these programs are intricately tied to how they operated and integrated with COPC principles. In that case, it might be useful to divide the findings section differently. In the present manuscript the findings are divided based on COPC principle into “Assessment of local needs and assets and use of evidence”, “Service Integration”, etc. While this is valuable, this division merges the stories of the four study sites together. I was forced to frequently jump back and forth between the “THE CHW PROGRAMMES AND THE COMMUNITIES THEY SERVE” section and the “COMMUNITY ORIENTATION OF THE CHW PROGRAMMES” section to understand which site has which characteristics. Again, this could be due to my ignorance about South Africa. But considering this is an international journal, I propose an alternative way of grouping the findings.

The findings could be grouped according to the sites. It would be very interesting to read “The experience from Sedibeng”, “The experience from Johannesburg”, “The experience from Tshwane”, and “The experience from Ehlanzeni” as four stories. This would help the reader understand the unique characteristics and history of each site, the rural/urban nature of the site, the funding mechanisms, the way the program was rolled out there, and the way it is practised there. This seems like a less confusing way to organize the findings. The authors may consider ignoring this idea if it leads to difficulties in organizing the findings within the story of each site.

The first paragraph of the discussion section (436-446) could be dissolved into the preceding section. If the authors so wish, it might be possible to merge the whole discussion section into the stories above by highlighting and comparing the relevant literature in the corresponding paragraphs within the story of each site. But if that’s too disruptive, the discussion reads fine as it is.

A few sentences/statements/observations struck me as lacking nuance:

44-46: Due to insufficient number of health workers, community health workers (CHW) are being deployed to provide health care services to under-served communities[3]

The reference doesn’t substantiate the claim that CHWs are being deployed due to insufficient number of health workers. CHWs are health workers. They have their own independent role that cannot be fulfilled by medical practitioners (nurses or doctors). The sentence as it stands makes it appear like CHWs are filling the vacuum of other health workers and provide some sort of weakened, diluted form of care than what would have been possible if there were a sufficient number of (other) health workers.

437-439: There was some evidence of resistance by community members to participate in the design and implementation of the health programmes, as the residents were more interested in housing than healthcare.

Housing *is* healthcare. This judgement on the residents is perhaps antithetical to COPC.

Copyediting errors:

The abstract includes a typo: “However, many residents were more concerned

about access to housing that health services delivery issue” - “that” should be “than”

440: “particularly were the CHWs” should be “particularly where the CHWs”

Reviewer #5: I very much enjoyed reading this research! CHWs are a cornerstone of global health, but we have too little insight on contemporary CHW experiences. I believe your research can make a contribution; however, revisions to the manuscript are needed.

Abstract

Discuss the timespan of the CHW program under study (following UHC efforts in 2011); otherwise, the reader does not know if this is a decades-long program or a new initiative.

Background/Introduction

Please provide a brief (1 well-cited paragraph) on the history of CHWs in global health - this has been a key strategy since the 1950s, gained prominence following Alma-Ata, and has had renewed energy through UHC. This will help connect your work and the experience in South Africa to global trends.

Further discussion of and citation of the huge CHW literature is vital for positioning your work.

You must define CHWs for your study. There is a very wide range of what is defined as a CHW globally, and we need a clear picture from the outset exactly what kind of group you are describing. In the background, you can describe the CHW program "on paper" - what is intended? How are CHWs selected/recruited? (We only learn in the Discussion (line 474) that communities have no involvement in selection.) Are CHWs paid? What is the scope of the role - what kinds of tasks? Is it full-time? Ideally, how are CHWs trained and supervised? We need to know all of this to then be able to understand program goals and compare to the reality of what is actually happening as presented in the results.

Methods

Note in Table 3 the number of participants in each focus group. (Also spell out Johannesburg in Table 3 instead of JHB.)

Please created a Table of participant demographics (gender, age, role) as much as possible.

Describe how focus group discussion participants were recruited.

Minor: Line 158 - were should be was

Clarify the difference between the in-depth interviews and key informant interviews. The descriptors in Table 3 and then the two sections describing them are unclear; please give more insight into how these were differentiated or collapse the categories together.

Clarify the consent processes for non-English speakers/readers, as you indicated that written consent was given prior to all research activities and that all instruments were in English.

Findings

Change section header to Results

Delete lines 188-190 - This introduction does not add to the section

Lines 203-206 seem out of place and are unclear. Please add further explanation, incorporate into other sections, or delete.

Line 227 - Define the timeframe; you indicate "several years," and we learn elsewhere that UHC CHW programs began in 2011 and the data were gathered from 2016-19. What time period are you referring to here?

Line 256 - Define RDP (only mention in paper)

Line 260 - Reword for judgement-neutral language, such as "residents faced challenges in maintaining medication protocols for chronic conditions."

Minor: Line 333 - Change make to makes

Line 380: Define OTL (only mention in paper)

Throughout the results section, it was challenging to follow the positionality of the respondents and data, since there are CHWs themselves, other health providers, other health system stakeholders, and community members included. Please clarify positions and differing views represented in your data based on role throughout sections.

Discussion

Overall, I felt like I was learning new things in the discussion that were not mentioned in the results. While the discussion should be a fresh analysis of your data, you should not be presenting new data in this section.

Line 442 - You mention medical doctors providing training for some CHWs. Make sure this is described in the results. Is this necessarily desirable/a programmatic goal?

Line 444 - You use the loaded phrase "volatile communities," though this notion is not presented in the data. What data support this conclusion? How does the CHW program help or is it hampered by a specific set of community conditions?

Line 452-454 - It is great that you loop back to some of the literature. However, in the Discussion, you must directly link or compare to your results. What is our takeaway comparison to the Brazil literature?

Line 458 - How frequently did CHWs rarely interact with facility-based providers? I know it is a qualitative dataset, but please give us a sense of proportionality in the results.

Conclusion

In both the Discussion and especially the Conclusion, you focus on lack of sustained funding as the downfall of CHW efforts. However, apart from the lapse of mobile devices, we do not get deep insight into this from your participants in the results. Make sure to focus the Discussion and Conclusion on the scope of your data. It is of course excellent for you to add your own analysis in these areas -- and funding cessation is clearly a key issue, but it was not the only issue raised in the data.

Reviewer #6: This is a relatively well written paper in simple and easy to read language.

I have only two (substantive) comments and a couple of editorial comets/observations

Substantive comments:

Page 5 – under data collection:

1. The authors say that data collection tools we in English: but the (data collection) team provided additional local language explanation/interpretation to the interview questions for participants who did not understand English. I would like the authors to comment on the views of possible varied explanations/interpretations of the interview questions (hence varied responses) due to lack of standardized translated interview guides. Related to this: I feel the authors need to acknowledge, include and reflect on lack of standardized translation of interview guided as part of the study limitations documented in page 20

2. The authors should include a brief section on recommendations/policy and practice implication of the study findings just before the conclusions

Minor editorial comments

1. The authors use both direct and indirect reporting styles. E.g. In the abstract “…… In this article, we explore to what extent a nation-wide CHW programme in South Africa is attuned to community needs, integrated into the healthcare….” ( direct reporting) and “…..e study was conducted in seven primary healthcare facilities located in semi-urban and rural …” (indirect reporting). There is need to stick to one style all through.

2. The authors refer to data in singular ( “ data was” ) in various places in the methods section. Data should be referred to in plural (“Data were”)

3. Page 12: “a mobile app” should be written in full as “ a mobile phone application”

4. Page 18 line 449 : “A review in low and middle income countries has found …..” . Should be: “A review of literature from low and middle income countries has found …”

6. PLOS authors have the option to publish the peer review history of their article (what does this mean?). If published, this will include your full peer review and any attached files.

**Do you want your identity to be public for this peer review?** For information about this choice, including consent withdrawal, please see our Privacy Policy.

Reviewer #1: No

Reviewer #2: No

Reviewer #3: No

Reviewer #4: **Yes: **Akshay S Dinesh

Reviewer #5: No

Reviewer #6: No

---

## [Editor Report · Decision Letter 1]

18 Jan 2023

Community-orientated primary health care: exploring the interface between community health worker programmes, the health system and communities in South Africa

PGPH-D-22-01143R1

Dear Mr Malatji,

We are pleased to inform you that your manuscript 'Community-orientated primary health care: exploring the interface between community health worker programmes, the health system and communities in South Africa' has been provisionally accepted for publication in PLOS Global Public Health.

I commend you on addressing all the comments from the peer reviewers. We had an unusually high number of reviewers for this paper and I appreciate you addressing the comments. The paper reads really well! Thank you

Best regards,

Shailendra Prasad, MD, MPH

Academic Editor
